# Inarticulate past: Similarity properties of the ice-climate system and their implications for paleo-records attribution

*Mikhail Y. Verbitsky*

Gen5 Group, LLC, Newton, MA, USA
UCLouvain, Earth and Life Institute, Louvain-la-Neuve, Belgium
Correspondence: Mikhaïl Verbitsky (verbitskys@gmail.com)

**Abstract.** Reconstruction and explanation of past climate evolution using proxy records is the essence of paleoclimatology. In this study, we use dimensional analysis of a dynamical model on orbital time-scales to recognize theoretical limits of such forensic inquiries. Specifically, we demonstrate that major past events could have been produced by physically unsimilar processes making the task of paleo-records attribution to a particular phenomenon to be fundamentally difficult, if not impossible. It also means that any future scenario may not have a unique cause and, in this sense, the orbital time-scale future may be to some extent less sensitive to specific terrestrial circumstances.

## Introduction

Interpretation of most prominent events of climate history such as the middle-Pleistocene transition (Ruddiman et al., 1986, Lisiecki and Raymo, 2005, Clark et al., 2021) has been an inspiration for several generations of climate modelers (see for a review Saltzman, 2002, Tziperman et al., 2006, Crucifix, 2013, Mitsui and Aihara, 2014, Paillard, 2015, Ashwin and Ditlevsen, 2015, Verbitsky et al, 2018, Willeit et al., 2019, Riechers et al, 2022). While specific physical mechanisms invoked to explain changing glacial rhythmicity vary, they all include slow changes of ocean-atmosphere governing parameters (e.g., Saltzman and Verbitsky, 1993; Raymo, 1997; Paillard and Parrenin, 2004) or glaciation parameters (Clark and Pollard, 1998). On a more general level, all these theories in fact assume slow changes in the intensities of positive (such as, for example, long-term variations in carbon dioxide concentration, e.g., Saltzman and Verbitsky, 1993) or negative (for example, regolith erosion, Clark and Pollard, 1998, or diminished role of the geothermal heat flux relative to the vertical temperature advection in growing ice sheets, Verbitsky and Crucifix, 2021) system feedbacks. Though all physical phenomena invoked are, indeed, real and may be plausible, the following question still remains unanswered: Is it possible to disambiguate the past and elevate a single "correct" theory? Answering this question is the goal of our study.

Indeed, this is the classical attribution challenge that has been successfully addressed in the context of another well-known problem of geophysics: the causality of the observed global warming. For this purpose, the most comprehensive space-resolving models have been employed to reproduce observed time-series under different conditions and to prove (or discredit) a candidate physical phenomenon (e.g., Stocker, 2014). Certainly, these models cannot be employed on extremely long orbital time-scales (10 – 100 kyr) due to computational constrains. In search for an alternative, we turn here to dimensional analysis. Historically, dimensional analysis and concepts of similarity have been used for studying physical phenomena, complementing even the most sophisticated computational tools and providing physical insight in situations where physical interpretation of the higher-complexity modeling results may be difficult. Here, on orbital timescales, when we retreat from physics-abundant space-resolving models to more conceptual dynamical models, dimensional analysis may be promoted from a supporting to a more prominent role.

Several key terms need to be introduced before we outline the structure of our paper. We will be using the definitions of similarities as they have been articulated by G. I. Barenblatt (2003). Suppose we have a physical phenomenon that is governed by $n$ physical parameters, $k$ parameters of which are parameters with independent dimensions. Then, according to $\pi$-theorem (Buckingham, 1914), the phenomenon can be described by $n$-$k$ adimensional similarity parameters $\pi_1, \pi_2, \ldots \pi_i \ldots, \pi_{n-k}$. We will consider two phenomena as being *physically similar* if they are described by identical similarity parameters $\pi_1, \pi_2, \ldots \pi_i \ldots, \pi_{n-k}$. The dimensionless time series of physically similar processes are also identical. If a similarity parameter $\pi_i$ can be excluded from the description of a physical process (a phenomenon becomes independent of it in the limit that $\pi_i$ tends to zero or infinity) we can talk about *complete similarity* of this physical process in this parameter: regardless of parameter's specific value, the process does not depend on it. And, finally, we may observe *incomplete similarity* when none of similarity parameters $\pi_1, \pi_2, \ldots \pi_i \ldots, \pi_{n-k}$ can be neglected even if they are too small (or too big), but the number of effective

parameters may still be reduced because a phenomenon depends not on actual values of similarity parameters but on
their products in some power degree (i.e., conglomerate similarity groups):
$\Pi_j = \left(\pi_1^{\alpha_j}\right)\left(\pi_2^{\beta_j}\right)...\left(\pi_i^{\lambda_j}\right)...\left(\pi_{n-k}^{\chi_j}\right)$ $(j = 1,2,...,l; l < n - k)$. Here $\alpha_j, \beta_j, ..., \lambda_j, ..., \chi_j$ are power degrees
of $\pi_1, \pi_2, ...\pi_i ..., \pi_{n-k}$ involved into $\Pi_j$ formulation.
We are now ready to proceed with the structure of our paper: (a) first, we will introduce our dynamical
model and describe major physical processes involved; (b) using dimensional analysis, we will define 8 similarity
parameters $\pi_1 - \pi_8$ that completely define model's behavior; (c) Since our system does not have a property of
complete similarity in any of individual parameters $\pi_1 - \pi_8$, we will attempt to discover incomplete similarity and
find conglomerate similarity groups. Unfortunately, there are no specific algorithms that can help us to determine
governing conglomerate similarity groups $\Pi_j$, if they indeed exist. Therefore, we will articulate such conglomerate
similarity groups based on observed system behavior; (d) we will then discuss implications of our findings for the
attribution challenge and illustrate our reasoning with a numerical experiment; (e) we will conclude our study with
some thoughts relating our results to the real-world climate system.
**Method**
For our experiments we employ the Verbitsky et al (2018), VCV18 thereafter, dynamical model of the ice-
climate system. It has been derived from the scaled mass- and heat-balance equations of the non-Newtonian ice
flow, i.e., equations (1) and (2), correspondingly, and combined with an energy-balance equation of the global
climate temperature (3):
$$\frac{dS}{dt} = \frac{4}{5}\zeta^{-1}S^{3/4}(a - \varepsilon F_S - \kappa\omega - c\theta) \qquad (1)$$
$$\frac{d\theta}{dt} = \zeta^{-1}S^{-1/4}(a - \varepsilon F_S - \kappa\omega)\{\alpha\omega + \beta[S - S_0] - \theta\} \qquad (2)$$
$$\frac{d\omega}{dt} = -\gamma[S - S_0] - \frac{\omega}{\tau} \qquad (3)$$
Here, $S$ (m$^2$) is the area of glaciation, $\theta$ ($^o$C) is the basal ice-sheet temperature, and $\omega$ ($^o$C) is the global temperature
of the ocean-atmosphere (rest of the climate) system. In deriving equations (1) and (2) we consider ice sheets in the
thin-boundary-layer approximation such that their inertial forces are negligible relative to stress gradients, and
motion equations with very high accuracy can be written in a quasi-static form. For such approximation, a
characteristic ice thickness $H$ is connected to ice area $S$ as $H = \zeta S^{1/4}$ where $\zeta$ (m$^{1/2}$) is a profile factor assumed to
be constant (Verbitsky and Chalikov, 1986, VCV18). Further, equation (1) represents global ice balance $\frac{d(HS)}{dt} =$
$AS$, where, again, $H = \zeta S^{1/4}$ and $A = a - \varepsilon F_S - \kappa\omega - c\theta$ is the surface mass influx. Equation (2) describes
vertical ice temperature advection with a time scale $H/(a - \varepsilon F_S - \kappa\omega)$, and equation (3) is the global energy-
balance equation. The parameter $a$ (m s$^{-1}$) is the snow precipitation rate; $F_S$ is normalized external forcing,
specifically, mid-July insolation at 65°N (Berger and Loutre, 1991) of the amplitude $\varepsilon$ (m s$^{-1}$) such that $\varepsilon F_S$ describes
ice ablation rate due to astronomical forcing; $\kappa\omega$ is the ice ablation rate representing the cumulative effect of the
global climate on ice-sheet mass balance; $c\theta$ represents ice discharge due to ice-sheet basal sliding; $\alpha\omega$ is basal
temperature response to global climate temperature change, $\beta[S - S_0]$ is basal temperature reaction to the changes
of ice geometry; $-\gamma[S - S_0]$ describes global temperature response to ice geometry changes (e.g., albedo); $\kappa$ (m s$^{-1}$
$^o$C$^{-1}$), $c$ (m s$^{-1}$ $^o$C$^{-1}$), $\alpha$ (adimensional), $\beta$ ($^o$C m$^{-2}$) and $\gamma$ ($^o$C m$^{-2}$ s$^{-1}$) are sensitivity coefficients; $S_0$ (m$^2$) is a reference
glaciation area; and $\tau$ (s) is the timescale for $\omega$. When orbitally forced, the model reproduced events of the last
million years reasonably well, except for the interglacial of 400 kyr ago (marine isotopic stage 11). The timing of all
other interglacials coincides with Past Interglacial Working Group of PAGES (2016) data (VCV18).
We will now focus on the most remarkable feature of the historical records - a period $P$ of climate response
to the astronomical forcing. Indeed, it is the change of the climate variability from the predominant period $P = 40$
kyr to the main periods of $P = 80\text{-}120$ kyr that makes the middle-Pleistocene transition so extraordinary. Though the
amplitude increase was considered, until recently, to be a necessary attribute of this transition, its presence in the
paleo-records is now questioned (Clark et al, 2021). We begin with the dimensional analysis of the VCV18 system
(1) – (3). Indeed, it has 11 governing parameters (including the amplitude $\varepsilon$ and the period $T$ of the external forcing).
If we choose $\varepsilon$, $T$ and $\gamma$ to be parameters with independent dimensions, then in accordance with $\pi$-theorem a period
of the system response can be fully described by 8 dimensionless similarity parameters $\pi_1 - \pi_8$ :

$$\pi_1 = \frac{\varepsilon}{a}, \pi_2 = \alpha, \pi_3 = \kappa\gamma\varepsilon T^3, \pi_4 = c\gamma\varepsilon T^3, \pi_5 = \frac{T}{\tau}, \pi_6 = \frac{\gamma T}{\beta}, \pi_7 = \frac{S_0}{\varepsilon^2 T^2}, \pi_8 = \frac{\varsigma}{\varepsilon^{1/2} T^{1/2}}, \text{ and}$$

$$P = T\Psi(\pi_1, \pi_2, \ldots, \pi_8) \tag{4}$$

The numerical experiments with the system (1) – (3) demonstrate that individual similarity parameters $\pi_1 - \pi_8$
cannot be discarded using simply "too big" or "too small" arguments. It means that the system (1) – (3) does not
have a property of complete similarity in any of individual parameters $\pi_1 - \pi_8$. At the same time, we observed
earlier (Verbitsky and Crucifix, 2020) that the period of the system (1) – (3) response to the obliquity forcing of
period $T$ is mostly governed by two dimensionless parameters: by the ratio of the astronomical forcing amplitude to
terrestrial ice sheet snow precipitation rate, $\varepsilon/a$, and by the adimensional $V$-number. The physical meaning of the $V$-
number in the orbital domain becomes most evident if we take a closer look into the structure of positive and
negative feedbacks as they appear in the system (1) – (3). The time-dependent negative feedback is proportional to
the ice sheet area size as $\beta(S - S_0)$. The coefficient $\beta$ is defined by thermodynamical properties of an ice sheet,
most importantly by the Peclet number, $Pe = \hat{A}H/k$, $\hat{A}$ is a characteristic mass influx, i.e., accumulation minus
ablation and $k$ is ice temperature diffusivity (VCV18, Verbitsky and Crucifix, 2021). This negative feedback acts on
ice-sheet mass balance with a vertical-advection time delay and is amplified by a sensitivity coefficient $c$ that
reflects the intensity of basal sliding. The time-dependent positive feedback is global temperature $\omega$. In the orbital
domain, $\tau \ll T$ ($\pi_5 \gg 1$), $\omega$ is approximately proportional to $-\gamma\tau(S - S_0)$. The global temperature acts on the ice-
sheet mass balance "instantly" as $\kappa\omega$ and with the vertical-advection time-delay as a component of basal temperature
conditions, $\alpha\omega c$. Thus, the $V$-number is emerging in the orbital domain as a ratio of amplitudes of time-dependent
positive and negative feedbacks.

$$V = \frac{\gamma\tau}{\beta c}(\alpha c + \kappa) \tag{5}$$

Specifically, when $V \sim 0.75$ and $\varepsilon/a \sim 1$, the system exhibits the obliquity-period doubling. When the positive
feedback and the obliquity forcing are less articulated, the system responds with the 40-kyr period. Thus, slow
changes of the $V$-number (for example, from $V = 0.5$ at $t = 3,000$ kyr ago to $V = 0.75$ at $t = 0$) and of the $\varepsilon/a$ ratio (for
example, from $\varepsilon/a = 0.3$ to $\varepsilon/a = 1.7$ over the same time span) produce a change in the ice-climate behavior similar
to the middle-Pleistocene transition.
We now notice that the $V$-number can be presented in terms of similarity parameters $\pi_1 - \pi_8$, specifically:

$$V = \frac{\gamma\tau}{\beta c}(\alpha c + \kappa) = \left(\pi_2 + \frac{\pi_3}{\pi_4}\right)\frac{\pi_6}{\pi_5} \tag{6}$$

We also experimentally established that the period-doubling sustains ($\Psi = 2$) if, under fixed $\varepsilon/a$ and $V$, the period
of the external forcing changes from let say $T = 35$ kyr to $T = 50$ kyr. It can only happen if in this domain similarity
parameters $\pi_7$ and $\pi_8$ make another conglomerate similarity group that does not depend on $T$, specifically $\frac{\pi_8^4}{\pi_7}$.
Thus, equation (4) can be written as

$$P = T\Psi\left(\pi_1, \frac{\pi_2\pi_6}{\pi_5}, \frac{\pi_3\pi_6}{\pi_4\pi_5}, \frac{\pi_8^4}{\pi_7}\right), \tag{7}$$

that is the pure case of incomplete similarity as we defined it above: none of the similarity parameters can be
neglected but instead of 8 governing parameters we have been able to migrate to 4 governing conglomerate
similarity groups. Finally we may notice that $\frac{\pi_8^4}{\pi_7} = \frac{H^4}{S_0^2} \ll 1$ for all large ice sheets. If, we set it to be constant, we
can re-write equation (7) in a more simple form as

$$P = T\Psi\left(\frac{\varepsilon}{a}, V\right) \tag{8}$$

151       Recognition of governing conglomerate similarity groups is important because it provides us with a powerful
insight: different combinations of similarity parameters $\pi_i$ may produce the same $V$-number, i.e., *physically*
*unsimilar processes* (formed by not identical $\pi_i$) *may cause the same outcome.* This observation is critical for our
attribution challenge. Certainly, precise disambiguation of historical records is always a difficult task because even
two physically similar processes having identical adimensional similarity parameters and demonstrating the same
behavior may have been produced by different values of physical parameters involved, unless these parameters are
physical constants or well defined. The situation becomes especially challenging when we deal with conglomerate
similarity groups because, as we just stated, the same results may be produced by not-identical similarity parameters
(physically unsimilar processes). This is the theoretical limit that we aspire to expose.
160       We will now apply our findings to the middle-Pleistocene transition. Since the physical interpretation of the
governing parameters incorporated in the conglomerate *V*-number is very straightforward, we may observe a similar
(in terms of the period-*P* bifurcation) system response to changes of a completely different physical nature. For
example, parameter $\beta$, as we have discussed above, defines intensity of the negative feedback and is formed as a
result of interplay between vertical ice advection, internal friction, and geothermal heat flux (VCV18). Increased
Peclet number of growing ice sheet diminishes the role of the geothermal heat flux and may reduce parameter $\beta$ thus
increasing the *V*-number. The same period-*P* bifurcation can also be caused, for example, by slow changes in the
parameter $\gamma$ that defines the intensity of the positive feedback and incorporates effects of the albedo change or other
atmospheric feedbacks. We solve equations (1) – (3) for these two cases. In both cases we invoke a global cooling
trend. In our first experiment (Fig. 1a), this trend is translated into reduction of $\beta$, i.e., weakening of the ice sheet
negative feedback, and corresponding increase of the *V*-number from $V = 0.5$ to $V = 0.75$. The increased
continentality of the climate (reduced intensity of the snowfall during colder climate) is accounted by the $\varepsilon/a$ ratio
increase from $\varepsilon/a = 0.3$ to $\varepsilon/a = 1.7$. In the second experiment (Fig. 1b), the *V*-number also evolves from $V = 0.5$ to $V$
$= 0.75$, but this time it is achieved by increased intensity of the positive feedback ($\gamma$). In both experiments, we used
mid-July insolation at 65°N (Berger and Loutre, 1991) for the last 3 million years as an astronomical forcing. The
millennial forcing is added to $\varepsilon F_S$ as a single sinusoid of 5 kyr period and doubled ($2\varepsilon$) amplitude. It is important to
note that in the first experiment (changing $a$ and $\beta$) only similarity parameters $\pi_1$ and $\pi_6$ are being changed, but in
the second experiment (changing $a$ and $\gamma$) the same changes of the *V*-number are caused by changing $\pi_1, \pi_3, \pi_4, \pi_6$.
It means that the processes involved in these two experiments are not physically similar. Though the time-series
produced in these two cases are obviously non-identical (see Fig. 1 inserts), we can observe that different physical
phenomena may produce the same changes in the conglomerate *V*-number and the same large-scale effect, i.e., the
period-doubling bifurcation at about 1 Myr ago.
182       We do not attempt here to fully reproduce paleo-records such as the Lisiecki and Raymo (2005) and a
discussion of whether a period doubling should be accompanied by the amplitude increase is outside of the current
paper's scope. We will just remark that the amplitude of the system response is the function of not just the period *P*
but also of the $\varepsilon/a$ ratio (Verbitsky and Crucifix, 2020) and, for example, less articulated continentality of colder
climates may explain diminished amplitude contrasts as it has been recently advocated by Clark et al (2021).
187       Indeed, as we have already indicated, we used mid-July insolation at 65°N for the last 3 million years as an
astronomical forcing. Apart from that, these examples may also serve as an illustration of some future scenarios of
the climate system behavior under post-industrial atmospheric carbon dioxide concentration reduction as implied by
Ridgwell and Hargreaves (2007). Again, regardless of the physical nature of the underlying dynamical system, it
exhibits 40-kyr rhythmicity of the first 1.5 million years of its evolution and consequent obliquity-period doubling.
This probable renaissance of ice-ages is different from the one envisioned by Talento and Ganapolski (2021) which
is based on the model tuned to the late Pleistocene (last 800 kyr) ice-volume data and thus postulates only 100-kyr-
period variability for the future.
**Conclusions**
The idea of the current presentation is simple but its implication may be important: If ice-climate system is defined
by conglomerate similarity groups, then we may be limited in our ability to disambiguate historical records and
different physical processes may produce same future scenarios. The latter is intriguing because since B. Saltzman
(1962) and E. Lorenz (1963) had discovered a hydrodynamic system's sensitivity to initial conditions, the concept of
deterministic chaos became a dominant concept of weather and climate theory. Our findings suggest that if we
consider orbital time scales and, instead of time series, focus on their more generalized attributes such as the period
of the system response to the astronomical forcing, we may observe that the behavior of these attributes may be, to
some extent, less sensitive to the physical nature of the terrestrial governing processes.
205       But do conglomerate similarity groups indeed govern the dynamics of the real orbital-scale climate system?
So far, these groups have been found only in our VCV18 low-order dynamical model, and although this model has
been explicitly derived from the conservation laws, our concept will remain hypothetical until it is supported by
empirical data. We speculate, though, that existing historical records may perhaps provide some support to our
theory. To evaluate the feasibility of a diagnostic approach, we entertain here a simple scaling exercise. Suppose
that an empirical time series, such as $\delta^{18}$O record, is created by a parent system (other than the VCV18) which is
controlled by $n$ physical parameters ($k$ of them having independent dimensions). If we choose the period of the
astronomical forcing $T$ to be among parameters with independent dimensions, then in accordance with the π-theorem
we have:
$$P = T\Psi(\pi_1, \pi_2, ..., \pi_{n-k})$$ (9)
The wavelet spectrum of the late Pleistocene $\delta^{18}$O variability in response to the precession (~20-kyr period) and
obliquity (~40-kyr period) forcing shows the dominance of 40-kyr and 80-kyr periods (Fig. 1c). If we are willing to
accept it as a hint of $\Psi = 2$ for $T = 20$ kyr and for $T = 40$ kyr, then, since some of the similarity parameters
$\pi_1, \pi_2, ..., \pi_{n-k}$ depend on $T$, the period-$T$ independence of $\Psi$ may only happen when $\pi_1, \pi_2, ..., \pi_{n-k}$ make
conglomerate $T$-independent groups. In other words, *period independence of the $\Psi$ function may be a fingerprint of*
*conglomerate similarity groups*. Indeed, the diagnostics of the $\Psi$ function may require much more sophisticated
instruments than our *ad hoc* reasoning, and the records will likely not explicitly reveal what the conglomerate
similarity groups look like; nevertheless, their mere existence would corroborate the idea of this paper.

**Competing interests:** The author declares that he has no conflict of interest.
**Acknowledgements:** The author is grateful to Michel Crucifix for multiple discussions related to this topic and to
Jeremy Bassis and anonymous reviewer for their helpful comments.

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

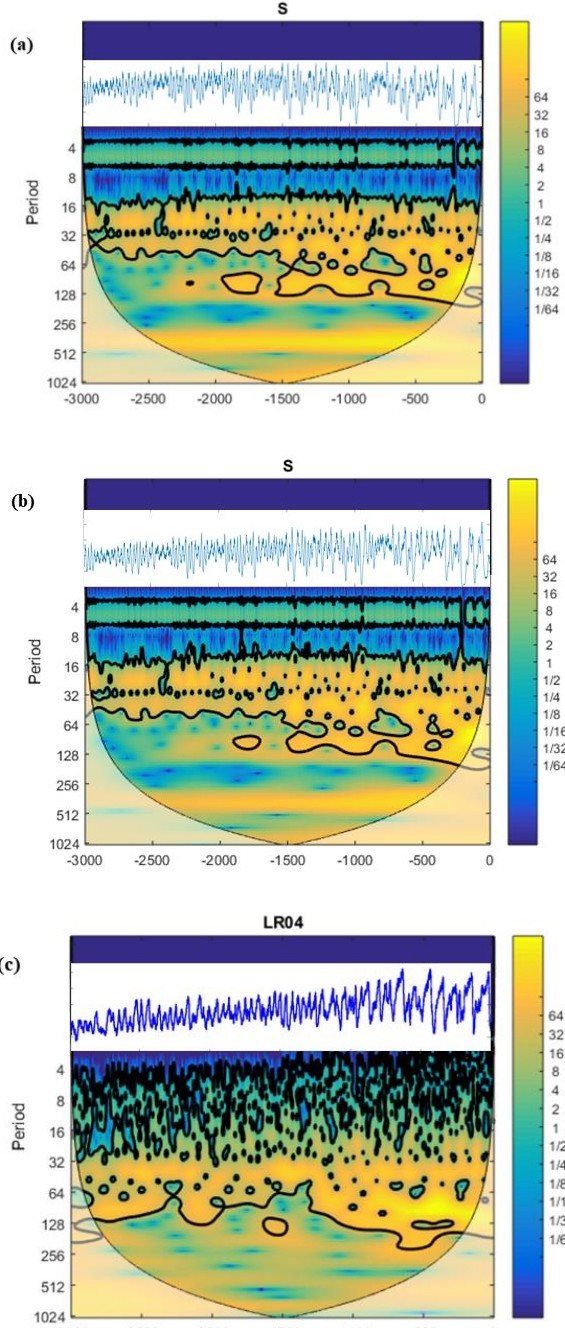

**Fig. 1** Ice-climate system response to a cooling trend presented as an evolution of wavelet spectra over 3 Myr for calculated ice-sheet glaciation area $S$ ($10^6 \, \text{km}^2$) – panels (a) and (b), and for the Lisiecki and Raymo (2005) benthic $\delta^{18}O$ record, panel (c). The $V$-number evolves from $V = 0.5$ to $V = 0.75$ due to weakening of the negative feedback (a) and due to intensified positive feedback (b). The vertical axis is the period (kyr), the horizontal axis is time (kyr before present). The color scale shows the continuous Morlet wavelet amplitude, the thick line indicates the peaks with 95 % confidence, and the shaded area indicates the cone of influence for wavelet transform. Inserts are corresponding time series.