# Peer review of "Inarticulate past: Similarity properties of the ice-climate system and their implications for paleo records attribution"

_Earth System Dynamics, 2021_

## Author Comment (AC1)

**Response to Anonymous Referee #1,**

Dear Referee 1, I am grateful for your comments and suggestions that will help me to improve the manuscript.

**General comments:** The author claims to have proved, through the use of dimensional analysis, that the ice-climate system possesses the property of incomplete similarity, concluding from there that a certain event at orbital time-scales could have different causes.
**Answer:** Your observation is correct.

**Comment:** Several aspects are not immediately clear to me: **(1)** Definitions for many crucial terms (incomplete similarity among them) are not properly presented in the manuscript and it is, thus, very difficult to follow the logic of the paper and evaluate the potential implications.
**Answer:** Definitions of some key terms such as physical similarity, complete similarity, and incomplete similarity were not conceived by the author but have been adopted from Barenblatt (2003). The goal of such adaptation was to simplify G. I. Barenblatt's rigorous formalism and to make it a bit more explicable to a wide audience. You comment clearly testifies that this goal is yet to be achieved.
**Action:** In the new version of the paper these definitions will be edited taking into account your multiple specific comments below.

**Comment: (2)** The core of the manuscript is the application of dimensional analysis to a simplified model, leading to the derivation of the adimensional parameter *V*-number. This has been done already in Verbitsky et al., 2018. I struggle to find what the added value of what is presented here is. In the current manuscript the author just modifies how the *V*-number changes in time (the *V*-number depends on several model parameters and it is, thus, possible to modify it by changing the values of those parameters in different ways). How is this a substantial knowledge contribution? Seems only an example and not enough material for a new publication.
**Answer:** This is, certainly, the most important comment, but before I respond to it (since the new version of the manuscript is not ready yet) I want you to become a bit more comfortable with some key terms. We will consider two phenomena as being *physically similar* if they are described by identical similarity parameters. For example, the air flow in a wind tunnel and a real flow are physically similar if they are both described by the same similarity parameters (like, e.g., Reynolds number) of the same value. The dimensionless time series of physically similar processes are identical. If a similarity parameter can be excluded from the description of a physical process (because, let say, it is negligibly small relative to other similarity parameters) we can talk about *complete similarity* of this physical process in this parameter: regardless of its specific value, the process doesn't depend on it. And finally, we may have *incomplete similarity*, when similarity parameters cannot be neglected even if they are small, but the number of effective parameters may still be reduced because a phenomenon depends not on absolute value of similarity parameters but on their ratios (in some power degree, i.e., conglomerate groups).

Now, we can talk about novelty. Yes, the *V*-number has been discovered experimentally (numerically) in the Verbitsky et al., 2018 (VCV18 thereafter) and has been discussed and explored extensively to demonstrate its defining role in VCV18 system dynamical properties, such as period doubling, scale invariance, *etc*. Finding incomplete similarity was not our initial target but when we figured out that VCV18 dynamics is defined by the ratio of its positive and negative feedbacks (the *V*-number) we then first become suspicious that incomplete similarity may be involved.

Indeed, the VCV18 system has 11 governing parameters, 3 of them are parameters with independent dimensions. It means that the VCV18 behavior can be fully described by 8 dimensionless similarity parameters $\pi_1 - \pi_8$ :

$$\pi_1 = \frac{\varepsilon}{a}, \pi_2 = \alpha, \pi_3 = \kappa\gamma\varepsilon T^3, \pi_4 = c\gamma\varepsilon T^3, \pi_5 = \frac{T}{\tau}, \pi_6 = \frac{\gamma T}{\beta}, \pi_7 = \frac{S_0}{\varepsilon^2 T^2}, \pi_8 = \frac{\varsigma}{\varepsilon^{1/2}T^{1/2}}$$

At the same time, we experimentally established that the period of the system response depends on smaller number of parameters, namely:

$$P = T\Psi(\pi_1, \Pi_1)$$
where

$$\Pi_1 = V = \left(\pi_2 + \frac{\pi_3}{\pi_4}\right)\frac{\pi_6}{\pi_5} = \frac{\gamma\tau}{\beta c}(\alpha c + \kappa)$$

It is the moment (after VCV18 has been already published) when we finally realized that we are dealing with incomplete similarity. This insight gives us much more than simply, as you say, "the *V*-number depends on several model parameters", in fact it is formed by several ***similarity parameters***. Therefore it provides us with additional powerful vision: different combinations of $\pi_i$ may produce the same *V*-number, i.e., physically unsimilar processes (formed by not identical $\pi_i$ ) may cause the same outcome.

To my knowledge this proposition is novel.

**Action:** We will include above reasoning in the paper.

**Comment: (3)** The author applies dimensional analysis to a 3-equations model of the ice sheets – climate system and makes some conclusions. How does the author then conclude that the real-world ice sheet- climate system also has the incomplete similarity property (whatever definition he is using)? I don't think it is possible to demonstrate, as the author claims, by the arguments shown here that the real-world system has the same a property as the simple model. There is some attempt to discuss this issue in the Conclusions, but the Abstract gives the misleading impression that a demonstration for the real-world case is provided in the paper.

**Answer:** This is definitely misinterpretation. The paper does not claim that incomplete similarity of the real-world ice-climate system has been proved. Instead, in the Conclusions (lines 146-149), the author is very explicit: "But is incomplete similarity of the global, orbital-scale, climate system real? So far, **this property has been found only in our VCV18 low-order dynamical model**, and although this model has been explicitly derived from the conservation laws, **the incomplete similarity of the ice-climate system will remain hypothetical until it is supported by empirical data**." Further, the author frames the answer to this question as a challenge for future research.

**Action:** If the Abstract contributed to such misinterpretation, its language will be revisited, otherwise no action is required.

**Comment: (4)** Even if the real-world system would have the incomplete similarity property (again, a concept not clearly defined in the text), which are the "theoretical limits for the use of paleo climate proxy records" that the author derives? Is the author implying that proxy records have no value for understanding millennial-scale fluctuations? Even if an event can be produced by several different

causes, why does this pose a theoretical limit for the use of proxy records? Which are the "far-reaching implications" that the author says might follow from the present study?

**Answer:** The goal of the paper is to demonstrate that disambiguation efforts, or in other words, attempts to attribute proxy records to a specific physical phenomenon may be fundamentally difficult if not impossible. Certainly, precise disambiguation of historical records is always a difficult task because even two *physically similar* processes having identical adimensional similarity parameters and demonstrating the same behavior may have been produced by different values of physical parameters involved, unless these parameters are physical constants or well defined. For example, suppose we have experimental measurements of an air flow made in a wind tunnel with a specific Reynolds number but the dimensions of the tunnel and air velocity data have been lost. Obviously, the same results can be produced by different tunnels having different dimensions and different flow velocities (air viscosity fortunately is the same) as long as the Reynolds number is the same. Thus, the task to attribute the results to a specific tunnel may be problematic. The situation becomes especially challenging when we deal with *incomplete similarity* because, as we discussed above, the same results may be produced by not-identical similarity parameters (physically unsimilar processes). This is the theoretical limit that we aspire to expose.

**Action:** The goal of the paper will be better articulated.

**Comment:** Overall, I find that the manuscript does not provide clear definitions for crucial terms (for example: physical similarity, complete and incomplete similarity) and is, therefore, difficult to read in its current form. Furthermore, given that a dimensional analysis has already been applied to the same (or a very similar) model in Verbitsky et al. 2018, how the material presented here constitutes a substantial new contribution is not at all evident. The differentiation from Verbitsky et al. 2018 must be made clearer and the new contributions highlighted. In its current form, the material here presented seems to be a simple illustration or example (V-value changing by changing the different model parameters that affect it).

**Action:** See responses and action items to comments (1) – (2).

**Particular comments:** I find that the introduction section must be substantially reformulated. Clear definitions and notations must be introduced. In addition, the author should clearly state the goal of the manuscript, which is missing in the current version.

**Action**: See response and action items to comments (1) - (4).

**Lines 9-10:** This sentence is not clear enough for an abstract (and repeats the word similarity 3 times). I would reformulate it and link to the following sentence. For example: Specifically, we demonstrate that major past events could have been produced by different physical processes and, therefore, the task of disambiguation of the historical paleo-records may be fundamentally difficult, if not impossible.

**Action:** The language will be revisited.

What is it there to disembogue? Please clarify

**Action**: See response and action item to comment (4)

**Lines 13-14:** Are you implying that glaciations are independent of orbital forcing? Is this paper implying we cannot forecast events at orbital time-scales? Any proofs?

**Answer:** There is no implication here that glaciations are independent of orbital forcing. There is no implication that we cannot forecast events at orbital time-scales. Instead, we demonstrate that orbitaltimescale predictions are, possibly, more stable than we might have thought – same results may be produced by multiple scenarios as long as they are based on the same *V*-number.
**Action:** This will be clarified

**Lines 21-24:** Please cite the work of Willeit et al. (2019).Willeit, M., Ganopolski, A., Calov, R., & Brovkin, V. (2019). Mid-Pleistocene transition in glacial cycles explained by declining CO2 and regolith removal. *Science Advances*, *5*(4),eaav7337.
**Action:** Thank you. It will be done

**Line 34:** change super long to extremely long. Also specify what does the author understand by orbital time scales in yrs.
**Action:** It will be done

**Line 40:** remove the word prophetic.
**Action:** It will be done

**Line 47:** asymptotic in which direction? What is the meaning of a parameter to disappear? Please, clarify, this is not clear.
**Action:** It will be clarified

**Lines 48-49:** what are the super indices mi, ni, qi? it is not defined. What does it mean pi in between brackets? The notation is not understandable, as it is not clearly defined. What's the meaning of conglomerate group?
**Action:** Thank you. I agree that notation here is not clear. It will be corrected.

**Line 52**: discover? did you prove the real-world system has this property? Or just your system of equations?
**Action:** See response to Comment 3.

**Line 55**: what is the definition of conglomerate *V*-number?
**Action:** The definition will be provided

**Line 60:** You said you already did this in VCV18, why do it again here?
**Action:** See response to Comment 2

**Lines 69-72:** Please, provide some extra description of the model. It is not obvious how terms to the power of 3/4 or 1/4 appear from the fundamental equations. A brief description of the main assumptions considered in the model derivation would be useful here. Also, mention if the model has been successfully validated against paleo data, or any indication of its ability to represent the real-life system.
**Action:** Additional description will be provided

**Line 76:** what is epsilon? It has units of velocity
**Action:** Clarification will be provided

**Line 78:** Are you referring now to the real-world dynamical system or the one of your equations?
**Action:** It will be clarified

**Line 88:** terrestrial ice sheet mass flux was earlier called snow precipitation rate. Please, use only one name.
**Action:** It will be done

**Line 93:** How is it derived that V represents that? Which are the positive/negative feedbacks? Below it says: beta is intensity of negative feedbacks, gamma is intensity of positive feedbacks. What is intensity??
**Action:** Although all these have been discussed in VCV18, I will briefly describe it here for clarity

**Line 95:** what is the meaning of slow here?
**Action:** Thank you. I agree that it needs clarification. It will be done.

**Line 100:** define the Peclet number
**Action:** It will be done

**Line 104:** "This feedback is applied directly to the ice sheet mass balance ($\gamma\tau\kappa$)" where is this derived?
**Action:** Although all these have been discussed in VCV18, I will briefly describe it here for clarity

**Line 105:** how is the $\gamma\tau\alpha c$ coefficient derived?
**Action:** Although all these have been discussed in VCV18, I will briefly describe it here for clarity

**Line 106:** what's the meaning of non-idealized?
**Answer:** "Non-idealized" here means that we solve pure system (1) – (3) without any parameters being neglected
**Action:** It will be clarified

**Line 107:** Please explain the meaning of "invoking a global cooling trend"
**Action:** It will be clarified

**Line 109:** define continentality
**Action:** It will be clarified

**Line 112:** what does the $m$ subscript represent?
**Answer:** "$m$" is for millennium
**Action:** No action is required

**Figure 1:** indicate that x-axis is time (kyr). Please also show the S time-series in each case.
**Action:** Thank you, time will be explained. All time-series will be provided as a supplement.

**Line 130:** The explanation of which astronomical forcing is used should be mentioned much earlier in the text.
**Action:** It will be done

**Line 146:** "But is incomplete similarity of the global, orbital-scale, climate system real?" I think there is no answer to this question in the manuscript, therefore, you can only focus in your model characteristics.
**Action:** This has been discussed in my response to Comment 3. No additional action is required

---

## Author Comment (AC2)

**Response to Anonymous Referee #2,**

Dear Referee 2, I am grateful for your comments and suggestions that will help me to improve the manuscript.

**Summary:** Overall, the goal of this manuscript is to demonstrate that a reduced order model of ice sheets exhibits incomplete similarity. I will be honest that I found this study to be hard to follow. I apologize to the authors in advance, if misunderstood what they did or said. Based on the difficult I had following the approach, I might not be the right person to review this manuscript. Nonetheless, my comments are below.

**Answer:** Thank you for your efforts. I am the one who should apologize for bringing to your attention a manuscript that is not explicable on its own. Yes, the goal of the paper, as you correctly observed, is "to demonstrate that a reduced-order model of ice sheets (*and climate* – MV) exhibits incomplete similarity". But this is only part of the goal. Most importantly, I wanted to demonstrate that, because of incomplete similarity, different combinations of similarity parameters (physically unsimilar processes) may lead to the same outcome.

I will explain this in my detailed answers below.

**Comment:** The basic system of equation is presented early on. It would make the manuscript much more accessible to provide an expanded description of the model and the physical interpretation of the parameters. For example, the parameter "a" is described as a snow precipitation rate. But the snowfall rate depends on the climate. Glacial cycles are known to be drier than interglacial cycles. And does the snowfall rate also include the melt rate? Or is that specified separately? Clearly, the melt rate has to depend on climate doesn't it? And then there are a host of "sensitivity coefficients". What do these physically represent and how would I measure them?

**Answer:** The model used in this study has been extensively described in Verbitsky et al, 2018 (VCV18 thereafter), and all questions you are raising above have been addressed there. I agree with you though that, for convenience of our readers, additional model description would be helpful.

**Action:** Additional model description will be provided

**Comment:** My next question arises from the assertion of incomplete similarity and description of what this means. Now I am vaguely familiar with similarity and incomplete similarity. The author's first assertion is that the period of the system only depends on two nondimensional numbers. Here it would be helpful to provide estimates of the physical magnitudes of each of the parameters based on whatever observations are available and to provide a physical interpretation for the "V-number" and why this controls the period. But I think my biggest question is I cannot follow the connection between the period doubling and incomplete similarity. The typical definition of complete similarity is usually that the similarity function becomes independent of some non-dimensional group in the limit that the non-dimensional number tends to zero or infinity. By contrast, the definition of incomplete similarity is that the similarity function does not become independent of the non-dimensional group as the group tends to zero or infinity. Instead, you end up with a scaling law where the scaling function becomes proportional to the non-dimensional group to some power. As the authors note, it is not usually possible to determine the scaling power by dimensional analysis alone. In the exposition (line 94), neither of the parameters tend to zero or infinity. The one parameter is 1 and the other is 0.75, neither of which can be considered large or small compared to one. So then the question: what does this have to do with incomplete similarity? When I have done calculations to determine incomplete similarity the goal has usually been to determine the scaling exponent, but I am uncertain if the authors even tried to find the

scaling exponent. I apologize to the authors if I misunderstood the analysis or their approach. Maybe I'm coming at it from the wrong direction.

**Answer:** Your understanding of complete and incomplete similarity is indeed correct, but, yes, in this study we approached incomplete similarity from a different direction. Finding of incomplete similarity was not our initial goal. We have been motivated to find physics responsible for period doubling bifurcation, and when we figured out that it is defined by the ratio of positive and negative feedbacks in the system (the *V*-number), we then first become suspicious that incomplete similarity may be involved.

Indeed, the VCV18 system has 11 governing parameters, 3 of them are parameters with independent dimensions. It means that the VCV18 behavior can be fully described by 8 dimensionless similarity parameters $\pi_1 - \pi_8$:

$$\pi_1 = \frac{\varepsilon}{a}, \pi_2 = \alpha, \pi_3 = \kappa\gamma\varepsilon T^3, \pi_4 = c\gamma\varepsilon T^3, \pi_5 = \frac{T}{\tau}, \pi_6 = \frac{\gamma T}{\beta}, \pi_7 = \frac{S_0}{\varepsilon^2 T^2}, \pi_8 = \frac{\varsigma}{\varepsilon^{1/2}T^{1/2}}$$

At the same time, we experimentally established that the period of the system response depends on smaller number of parameters, namely:

$$P = T\Psi(\pi_1, \Pi_1)$$
where

$$\Pi_1 = V = \left(\pi_2 + \frac{\pi_3}{\pi_4}\right)\frac{\pi_6}{\pi_5} = \frac{\gamma\tau}{\beta c}(\alpha c + \kappa)$$

It is the moment when we finally realized that we are dealing with incomplete similarity. This insight provides us with additional powerful vision: different combinations of $\pi_i$ may produce the same *V*-number, i.e., physically unsimilar processes (formed by not identical $\pi_i$) may cause the same outcome. To my knowledge this proposition is novel.

**Action:** We will include above reasoning in the paper.

**Comment:** I also did not understand the figures provided. The x-axis and colorer aren't labeled and the y-axis doesn't have units. What are we supposed to see here?

**Answer:** The horizontal axis is time (kyr before present), the vertical axis is the period of the system response (kyr), the color scale shows the continuous Morlet wavelet amplitude.

**Action:** The figure legend will be updated.

**Comment:** There is another subtle issue with the analysis which is that it is always difficult to determine if the behavior of a mathematical model is a feature of the simplifications of the mathematical model or is common to the more nuanced physics that is more representative of the physical system. Here it is unclear if the authors are claiming that their simplified model obeys incomplete similarity or if the general ice-climate system obeys incomplete similarity.

**Answer:** In this regard, the author is very explicit - see Conclusions (lines 146-149): "But is incomplete similarity of the global, orbital-scale, climate system real? So far, **this property has been found only in our VCV18 low-order dynamical model**, and although this model has been explicitly derived from the conservation laws, the **incomplete similarity of the ice-climate system will remain hypothetical until it is supported by empirical data**." Further, the author frames the answer to this question as a challenge for future research.

**Action:** If the Abstract contributed to such misinterpretation, its language will be revisited, otherwise no action is required.

---

## Author Response (AR3)

Dear Prof. Levermann,

Thank you for your acceptance decision.

It has been, indeed, a challenge to talk about incomplete similarity without mentioning it, but I did my best. I still say "incomplete similarity" once or twice to be consistent with Barenblatt's "gold-standard" definition, but otherwise I emphasize conglomerate-groups aspect of it and the absence of physical similarity. I have also changed the title to "Inarticulate past: Similarity properties of the ice-climate system and their implications for paleo-records attribution". I think it makes better presentation of the new accents.

Thank you again,

Mikhail Verbitsky